# Characterization, Communication, and Management of Uncertainty in Tuna Fisheries

**Gorka Merino [1],\*, Hilario Murua [2] , Josu Santiago [1], Haritz Arrizabalaga [1] and Victor Restrepo [2]**

1   AZTI, Marine Research, Basque Research and Technology Alliance (BRTA), Herrera Kaia, Portualdea z/g, 20110 Pasaia, Spain; jsantiago@azti.es (J.S.); harri@azti.es (H.A.)
2   International Seafood Sustainability Foundation, Washington, DC 20005, USA; hmurua@iss-foundation.org (H.M.); vrestrepo@iss-foundation.org (V.R.)
\*   Correspondence: gmerino@azti.es; Tel.: +34-667-174-456

**Abstract:** Tunas sustain important fisheries that face sustainability challenges worldwide, including the uncertainty inherent to natural systems. The Kobe process aims at harmonizing the scientific advice and management recommendations in tuna regional fisheries management organizations (RFMOs) toward supporting the sustainable exploitation of tunas globally. In this context, we review the similarities and differences among tuna RFMOs, focusing on stock assessment methodologies, use of information, characterization of uncertainty and communication of advice. Also, under the Kobe process, tuna RFMOs have committed to a path of adopting harvest strategies (HSs), also known as management procedures (MPs), which are the series of actions undertaken to monitor the stock, make management decisions, and implement the management measures. The adoption of HSs for tuna stocks is supported by Management Strategy Evaluation (MSE), which is considered the most appropriate way to assess the consequences of uncertainty for achieving fisheries management goals. Overall, notable progress has been made in achieving some of the Kobe objectives, but there are still some aspects that are inconsistent and need to be agreed upon, due to their management implications. First, not all RFMOs report on stock status based on maximum sustainable yield (MSY) as a reference. Instead, some use depletion level to represent the available fish biomass. Also, the definition of overexploited is not common in all oceans. Finally, very few stock assessments characterize all major sources of uncertainty inherent to fisheries. With regards to HSs, two different approaches are being followed: One is designed to adopt an automatic decision rule once the stock status and management quantities have been agreed upon (harvest control rules (HCRs), not strictly an HS) and the other aims at adopting all the components of HSs (data, use of information and decision rule).

**Keywords:** fisheries management; uncertainty; management strategy evaluation; tunas; RFMO; scientific advice; stock assessment

## 1. Introduction

Tunas sustain some the of world's most valuable fisheries and dominate marine ecosystems worldwide [1]. Tuna and tuna-like fish catch amounted 7.5 M tons in 2016 (9% of the global marine fisheries production) [2], which are worth more than EUR 32.54 billion to the annual global economy [3]. The seven most important tuna commercial species (albacore, ALB-*Thunnus alalunga*; Atlantic bluefin, BFT-*Thunnus thynnus*; Pacific bluefin, PBT-*Thunnus orientalis*; southern bluefin, SBT-*Thunnus maccoyii*; bigeye, BET-*Thunnus obesus*; yellowfin, YFT-*Thunnus albacares*; and skipjack, SKJ-*Katsuwonus pelamis*) account for 4.9 M tons and their 23 stocks sustain important industrial and artisanal fisheries [4]. Five regional fisheries management organizations (RFMOs) are responsible for the management of tuna

stocks: The Commission for the Conservation of Southern Bluefin Tuna (CCSBT), the Inter-American Tropical Tuna Commission (IATTC), the International Commission for the Conservation of Atlantic Tunas (ICCAT), the Indian Ocean Tuna Commission (IOTC), and the Western Central Pacific Fisheries Commission (WCPFC). Guided by their own conventions and consistent with the principles of the United Nations (UN) Fish Stocks Agreement [5] and the Food and Agriculture Organization (FAO) Code of Conduct for Responsible Fisheries [6], all five RFMOs are responsible for maintaining tuna stocks at sustainable levels. Over the last decades, tuna fisheries have intensified and expanded worldwide and global catch has steadily grown [7], which has placed all tuna RFMOs facing similar sustainability challenges [8].

The sustainability of fisheries is determined by the balance between the amount of biomass harvested and the resilience of fish stocks to harvesting. Tuna RFMOs generally aim at achieving the maximum sustainable yield (MSY), an equilibrium point at which the capacity of fish stocks to replace the removed biomass is maximized, and therefore, fisheries' long-term average catch is maximized too. The levels of catch and fishing mortality that can be sustained by fish stocks and their historical exploitation levels are estimated through stock assessment. Stock assessment is the process of collecting and analyzing stocks' biological and statistical information from fisheries to determine the changes in the abundance of fishery stocks in response to fishing, and, to the extent possible, to predict future trends in stock abundance (FAO Glossary of Terms). Fisheries stock assessments consist of fitting the available fishery and biological information into fish population dynamics equations using specifically tailored models and computer software.

A model can never describe a system with certainty [9], and thus, fish stock assessments are subject to uncertainty. In general, uncertainty can be classified into (i) systemic or structural uncertainty and (ii) aleatoric or statistical uncertainty [10]. Systemic or structural uncertainty is attributed to things that could potentially be known but they are not known in practice. This may be because a measurement is not accurate enough (observation error) or because models neglect certain effects (model error). Statistical uncertainty is representative of unknowns that differ each time we run the same experiment (process error), and it is acknowledged will not be possible to determine sufficiently to eliminate deviations from predicted values. Uncertainty quantification and characterization tries to determine how likely certain outcomes are if some aspects of the system are not exactly known [11,12]. Also, uncertainty quantification intends to work toward reducing the systemic uncertainties to statistical uncertainties, which can be narrowed down with statistical methods and characterized using probabilistic distributions [10,12,13]. In other words, uncertainty characterization aims to reducing bias and produce outputs that normally (or log-normally) fluctuate around a central tendency. Characterizing uncertainty in fisheries models is important because it helps provide a measure of the precision of the system dynamics, which is linked to the risk of adopting alternative management measures. In this sense, the higher uncertainty means a lower precision and a larger risk of not achieving sustainability.

A number of authors have categorized the errors that cause uncertainty and risk in fisheries [14–16]. In brief, three major sources of error are responsible for the uncertainty inherent to fisheries stock assessments [16]: (i) observation errors, directly linked to the quality and quantity of data used; (ii) model errors, due to the limited ability of models to reproduce population dynamic patterns; and (iii) process errors, due to the lack of understanding of the biological processes underlying fish stock dynamics. In this work, we review how each of the systemic and statistical uncertainties is characterized and quantified in tuna stock assessments with emphasis on similarities and differences across tuna RFMOs.

Stock assessments, including the characterization of uncertainty, are the cornerstone of the advice communicated by scientific committees at RFMOs. Initiatives like the Kobe framework [17] have aimed at providing consistency of advice across tuna RFMOs. However, the way that these are communicated to management bodies is still not homogenous. In this study, we review how the scientific advice is provided in the five tuna RFMOs with a focus on how uncertainty is communicated and its implications for management.

The uncertainty in fisheries and scientific advice, together with two international agreements—the UN Fish Stocks Agreement [5], and the FAO Code of Conduct for Responsible Fisheries [6]—provide the foundations of the precautionary approach (PA) to fisheries management, which aims at improving the management of fish resources by exercising prudent foresight to avoid unacceptable or undesirable situations, taking into account that changes in fisheries systems are not well understood and are only slowly reversible [18]. The PA seeks to protect fish stocks from fishing practices that may put their long-term viability in jeopardy despite the many unknowns on stocks biology, response to fishing, or exact state of exploitation [19]. Also, the PA requires that undesirable outcomes be anticipated and measures taken to reduce the probability of them occurring [20].

The PA calls fisheries management institutions to address uncertainty by determining the status of fish stocks relative to target, threshold, and limit reference points (RPs), to predict the outcomes of management alternatives for reaching the targets and avoiding the limits, and to characterize the uncertainty in both cases. Limit reference points (LRPs) are benchmarks that should be avoided with substantial probability, according to a given set of management objectives. They indicate the limit beyond which the state of a fishery and/or a resource is not considered desirable, and remedial management action is required to allow recovery. In contrast, a target reference point (TRP) is a benchmark that should be achieved on average, according to a given set of management objectives. It corresponds to a state of a fishery and/or resource that is considered desirable [21]. The PA also recommends that LRPs and TRPs are used in combination with precautionary or threshold RPs to determine what actions to be taken to avoid reaching the LRPs. A trigger or threshold is a level of biomass or fishing mortality rate between the LRP and TRP that can serve as a "red flag" and may trigger specific management actions designed to reduce fishing mortality.

One way to address the uncertainty in fisheries is the adoption of harvest strategies (HSs), also known as management procedures (MPs), that aim at both achieving a low probability of breaching safe biological limits (LRPs) and providing high average long-term catch and fisheries performance [22]. HSs are the systematic series of agreed human actions undertaken to monitor the stock, assess its state, make management decisions, and implement the management advice. A HS can be designed to specify changes to the total allowable catch (TAC), or any other measure, based on updated monitoring data and methods of analysis. Adopting an HS requires specifying the management objectives (probabilities, time frames and risk), reference points (TRPs and LRPs), performance indicators to monitor how effective the management measure is, the data and methods of analysis to determine the current state of the resource, and a decision rule (or harvest control rule, HCR) based upon the estimated state of the stock (including fishery indicators).

Management strategy evaluation (MSE) is considered the most appropriate way to assess the consequences of uncertainty for achieving fisheries management goals [23], and it is contributing to the increased robustness of tuna assessment and to supporting the adoption of HSs across RFMOs. MSE can be used to quantify the impacts of the uncertainty associated with current management systems and to identify "realizable" performance, which can be achieved given the quality of the data available and the types of uncertainties that are inherent to each fishery [23]. The five tuna RFMOs have carried out some type of MSE work, including consultation on management objectives, characterization of uncertainty of stocks' dynamics and observation, and evaluation of HCR and HS [22]. In the last section of this manuscript we review the state of play of the MSE process and the prospects for adopting HSs and HCRs for the key tuna stocks.

In summary, in this work we review the methods used in tuna stock assessments with emphasis on how uncertainty is characterized and communicated to management bodies. We also discuss how RFMOs are managing the uncertainty inherent to fisheries, with specific focus on the MSE processes and the adoption and evaluation of HSs for the most important tuna stocks. We do this with the aim of contributing to a harmonized global stock assessment and management framework for tuna stocks. We note that this review includes stock assessments carried out until 31 August 2020.

## 2. Characterization of Uncertainty in Tuna Stock Assessments

When making predictions with computer models as done in stock assessments, two types of uncertainties are encountered: systemic or structural and aleatoric or statistical [10].

1. Systemic or structural uncertainty is the result of model uncertainty and input uncertainty.

Model uncertainty: A model is said to display uncertainty when we are uncertain about the true relationship between inputs and outputs within a model [9]. One strategy to characterize this uncertainty is to use alternative stock assessment methods to add contrast to model estimates. The stock assessment models currently in use in tuna RFMOs range from the relatively simple catch-based models to more sophisticated age-/size-structured and fully integrated and flexible model platforms (Table 1, see Appendix A for a short description of models). However, it is only in ICCAT where the results of structurally different models are combined to provide scientific advice. While in the IATTC, CCSBT, WCPFC, and IOTC one fully integrated stock assessment model is used to provide scientific advice (Stock Synthesis [24], Multifan-CL [25], or one ad hoc model for Southern bluefin); in ICCAT, the results of fully integrated models are combined with simpler biomass production or age-structured models. For example, the stock status for Atlantic yellowfin was estimated averaging the results of three stock assessment models (Stock Synthesis, Just Another Bayesian Biomass Assessment (JABBA), a Bayesian biomass dynamic model [26], and mpb, a biomass-based stock assessment model [27]). In the case of west Atlantic bluefin, the results of a virtual population analysis (VPA/ADAPT [28]) and stock synthesis were combined. To sum up, model uncertainty is only characterized in two Atlantic stocks from the 23 most important tuna stocks.

**Table 1.** Models used in the latest assessments of the 23 most commercially relevant tuna stocks classified in relation to regional fisheries management organization (RFMO), stock name, year of the assessment, and method. A brief summary of each model is provided in Appendix A. ASPIC (A Surplus-Production model Incorporating Covariates), mpb (Biomass production model), JABBA (Just Another Bayesian Biomass Assessment), ASPM (Age Structured Production Model), VPA (Virtual Population Analysis), MFCL (Multifan-CL), SS (Stock Synthesis), SBT (Southern Bluefin Tuna model).

| RFMO | Stockname | Year | Indicators | Catch<br>Martell and Froese | Biomass Production<br>ASPIC | mpb | JABBA | Age/Size-Based<br>VPA | Fully Integrated<br>MFCL | SS | SBT |
|---|---|---|---|---|---|---|---|---|---|---|---|
| ICCAT | Atlantic yellowfin | 2019 | | | | • | • | | | • | |
| | Atlantic bigeye | 2018 | | | | | | | | • | |
| | East Atlantic skipjack | 2014 | • | • | | | | | | | |
| | West Atlantic skipjack | 2014 | | | • | | | | | | |
| | North Atlantic albacore | 2020 | | | | • | | | | | |
| | South Atlantic albacore | 2020 | | | | | • | | | | |
| | Mediterranean albacore | 2017 | | | | | • | | | | |
| | East Atlantic bluefin | 2020 | | | | | | • | | | |
| | West Atlantic bluefin | 2020 | | | | | | • | | • | |
| IOTC | Indian Ocean albacore | 2019 | | | | | | | | • | |
| | Indian Ocean bigeye | 2019 | | | | | | | | • | |
| | Indian Ocean yellowfin | 2018 | | | | | | | | • | |
| | Indian Ocean skipjack | 2017 | | | | | | | | • | |
| CCSBT | Southern bluefin | 2017 | | | | | | | | | • |
| IATTC | East Pacific yellowfin | 2020 | | | | | | | | • | |
| | East Pacific bigeye | 2020 | | | | | | | | • | |
| | East Pacific skipjack | 2019 | • | | | | | | | | |
| WCPFC | Pacific bigeye | 2018 | | | | | | | • | | |
| | Pacific yellowfin | 2017 | | | | | | | • | | |
| | South Pacific albacore | 2018 | | | | | | | • | | |
| | Pacific skipjack | 2019 | | | | | | | • | | |
| ISC | Pacific bluefin | 2018 | | | | | | | | • | |
| | North Pacific albacore | 2020 | | | | | | | | • | |

Input uncertainty: This arises when there is no certainty about the input parameters or the quality of the information. In fisheries science, it is attributed to the lack of biological knowledge about key processes and to inaccurate or incomplete data sources. This uncertainty is often characterized by developing model configurations using different model inputs (parameters and data) to characterize the potential impact on model estimates from knowledge gaps in key biological and fishery processes. These alternative configurations are often combined in factorial designs, which are known as uncertainty grids of models (Table 2). One way to characterize uncertainty of biological (or other) processes is to build model configurations with a range of fixed values, so that it is likely that the "true" value of the parameter is within the chosen range. For example, in fisheries one of the most difficult parameters to estimate is the steepness of the stock recruitment relationship, which is the fraction of recruitment from an unexploited population obtained when the spawning stock biomass (SSB) is at 20% of its unexploited level [29]. Therefore, it is not unexpected that steepness is used as a factor of uncertainty grids in tuna stock assessments [30] (Table 2), generally using fixed values between 0.65 and 1.

Other processes that are characterized as factors of uncertainty are natural mortality, growth, shape of selectivity curve, tag mortality, tag mixing period, selectivity, maturity, tag data overdispersion, and fecundity (included in the Psi parameter in the Southern bluefin tuna assessments).

Another source of input uncertainty is attributed to inaccurate or incomplete data. Generally, four types of data streams are used in stock assessments: catch, catch per unit of effort (CPUE) or other abundance indices, size frequency data, and tagging data. In some cases, data can follow opposing trends and inform the model in contradictory ways [31]. Often, different data sets are used to characterize the uncertainty in the data sources (Table 2). For example, in the West and Central Pacific stocks size information is downweighed with three options to modulate their influence. In the Indian Ocean stocks tagging information is considered uncertain, and therefore, two or three options for tagging data are used (weighted, downweighed, or omitted). With regards to the CPUE (or other abundance indices), the information of different indices is combined by weighting, adding, or omitting indices within the uncertainty grid.

In summary, all tuna RFMOs characterize input uncertainty using uncertainty grids with different factors as options for parameters and influence of data. Note that the IATTC grids are not exactly factorial grids but a combination of scenarios [32,33]. The largest grid is explored in the CCSBT where 432 models are used to characterize structural uncertainty. Also note that the stock assessments of Pacific bluefin (IATTC-ISC) and North Pacific albacore (WCPFC-ISC) do not use the uncertainty grid approach to evaluate stock status, and therefore, they are not included in Table 2.

In addition to the model grids, uncertainty is also explored through sensitivity runs, which are not included in the management advice but allow exploring the influence of the modeling choices made and the potential outcome of the assessment should these have been different. In general, sensitivity runs are used to support or complement the outcome of the assessment, but these results are not used to calculate the stock status or reference points for scientific advice.

2. Statistical uncertainty: This is mostly attributed to unpredictable variability in the abundance indices and recruitment deviations from model fits (process error). Statistical uncertainty can be characterized using statistical techniques such as the variance–covariance matrix, bootstrapping, delta or Monte Carlo methods, or others that assume that the deviations from a central tendency of the model outputs follow a probabilistic distribution.

**Table 2.** Number of options considered for each factor in the systemic uncertainty grids when using fully integrated stock assessment models. Sources: [34–42].

| | Model Inputs | ICCAT | | | IOTC | | | CCSBT | WCPFC | | | IATTC | |
|---|---|---|---|---|---|---|---|---|---|---|---|---|---|
| | | WA-BFT | A-BET | A-YFT | IO-BET | IO-YFT | IO-SKJ | SBT | P-BET | P-YFT | SP-ALB | EPO-BET | EPO-YFT |
| Parameters | Steepness | 1 | 3 | 0 | 3 | 3 | 3 | 3 | 3 | 3 | 3 | 4 | 4 |
| | Growth | 1 | 1 | 1 | 1 | 1 | 1 | 1 | 2 | 1 | 2 | 2 | 2 |
| | sigmaR | 1 | 3 | 1 | 1 | 1 | 1 | 1 | 1 | 1 | 1 | 1 | 1 |
| | Natural mortality | 1 | 2 | 1 | 1 | 1 | 2 | 12 ‡ | 1 | 1 | 2 | 2 | 1 |
| | Maturity | 2 | 1 | 1 | 1 | 1 | 1 | 1 | 1 | 1 | 1 | 1 | 1 |
| | tag mortality | 1 | 1 | 1 | 1 | 2 | 2 | 1 | 1 | 1 | 1 | 1 | 1 |
| | tag mixing period | 1 | 1 | 1 | 1 | 1 | 2 | 1 | 1 | 2 | 1 | 1 | 1 |
| | Fecundity (Psi) † | 1 | 1 | 1 | 1 | 1 | 1 | 3 | 1 | 1 | 1 | 1 | 1 |
| | Selectivity | 1 | 1 | 1 | 1 | 1 | 1 | 1 | 1 | 1 | 1 | 2 | 2 |
| | Recruitment regime | 1 | 1 | 1 | 1 | 1 | 1 | 1 | 1 | 1 | 1 | 2 | 1 |
| | Catchability | 1 | 1 | 1 | 1 | 1 | 1 | 1 | 1 | 1 | 1 | 1 | 3 |
| | tag data overdispersion | 1 | 1 | 1 | 1 | 1 | 1 | 1 | 2 | 2 | 1 | 1 | 1 |
| Data | weight size data | 1 | 1 | 1 | 1 | 1 | 1 | 1 | 3 | 3 | 3 | 2 | 1 |
| | weight tagging | 1 | 1 | 1 | 2 | 2 | 2 § | 1 | 1 | 1 | 1 | 1 | 1 |
| | weight CPUE | 1 | 1 | 2 | 1 | 2 | 1 | 4 | 1 | 1 | 2 | 2 | 1 |
| | regional structure | 1 | 1 | 1 | 1 | 1 | 1 | 1 | 2 | 2 | 1 | 1 | 1 |
| | Total | 2 | 18 | 2 | 6 | 24 | 48 | 432 | 72 | 72 | 72 | 44 | 48 |

(†) Psi is the power parameter on fecundity for the allometric relationship between fecundity and reproductive success. (‡) Four options for M at age 0, and three options for M at age 10. (§) Two tagging program options. Note that East Pacific Ocean assessments are not factorial grids. Also note that this table only contains the scenarios considered to provide scientific advice and excludes the scenarios used as sensitivity runs. WA stands for Western Atlantic, A for Atlantic, IO for Indian Ocean, P for Pacific, SP for Southern Pacific and EPO for Eastern Pacific Ocean.

The statistical uncertainty in model outputs is addressed by computing their confidence intervals. In general, variability is associated with data inputs such as abundance indices and deviations from the estimated recruitment. In the diagnostics of stock assessment fits, it is checked that deviations from fits to input data (abundance indices, length–frequency, tags, and catch data) and recruitment deviates can be characterized as probabilistic distributions, generally normal or log-normal. Once this is confirmed, confidence intervals of the output parameters are estimated using statistical techniques such as the delta method, likelihood profiles, bootstrapping, or alike. In the less computationally demanding models (biomass dynamic production) the bootstrapping is preferred [43]. First, a coefficient of variation is assigned to the abundance indices and the time series are replicated using random mean-square error, which are fitted in the same model configuration. Bootstrapping allows producing model outputs in the form of posterior distributions from which summary statistics can be obtained (e.g., mean, median, standard deviation, and confidence intervals). This is the common practice in the Atlantic stocks' assessments that use simpler models.

When the fully integrated models are used, it is often implausible to produce bootstraps due to time constrains within the assessment sessions (especially when systemic uncertainty is characterized using uncertainty grids). In these cases, the Hessian matrix can be computed from a base or reference case model to obtain estimates of the variance–covariance matrices, which are used in combination with the delta method [44] to estimate confidence intervals without needing to replicate the model fits. This method has been used to estimate statistics of model output parameters in the case of Indian Ocean albacore and all tuna stocks in the Pacific (IATTC and WCPFC). Since 2018, in the ICCAT and IOTC a delta–multivariate log-normal estimator [45,46] has been used to characterize statistical uncertainty from the model options of structural uncertainty grids. This method infers within-model uncertainty from maximum likelihood estimates, standard errors, and the correlation of model estimates [45], and it has the advantage of calculating probability coefficients in a few seconds of model computation, which makes it a promising method when dealing with large uncertainty grids run in time-limited stock assessment sessions.

One of the tasks of stock assessment is to predict future trends in stock abundance for alternative management measures. This is done in different ways across tuna RFMOs: In ICCAT and IOTC, the preferred option is to project the agreed upon stock assessment model or grid of models for a relatively wide range of catch levels, including the most recent catch. The projections for West Pacific stocks are set up to project current levels of fishing mortality (or catch). In the case of Eastern Pacific, projections are configured with the fishing mortality relative to MSY. In general, the projection period should be consistent with the lifespan of fish stocks. For most tunas species, 10–15-year projections are preferred, which allow for illustrating the impacts on fish abundance for the mid/long term. In the case of the latest assessment of Atlantic bluefin stocks, because of the large uncertainty in model estimates only a short-term projection was produced [47]. In the case of Southern bluefin, the long-term recovery is pursued, and projections are made for 25 years, but this is a special case due to the long life expectancy of bluefin tuna species compared to other tunas.

## 3. Communication of Uncertainty in Tuna RFMOs

The scientific advice provided to tuna RFMOs is based on the output of stock assessments. The Joint Meeting of Tuna RFMOs [17] agreed to harmonize the presentation of stock assessment results in order to improve the mutual understanding of scientists and managers in relation to the stock status determination, management advice, and the uncertainties inherent to both. The agreed upon presentation of stock assessment results was the Kobe plot (Figure 1a), a four-quadrant plot that frames the status of fish stocks in two dimensions: fishing mortality and biomass (or SSB), both relative to their MSY values. This allows reporting of stock status in four categories: (1) Green: Not overfished ($B > B_{MSY}$) and not undergoing overfishing ($F < F_{MSY}$); (2) Orange (or upper-right yellow) area: Not overfished ($B > B_{MSY}$) but undergoing overfishing ($F > F_{MSY}$); (3) Yellow (lower-left): Overfished ($B < B_{MSY}$) but not undergoing overfishing ($F < F_{MSY}$); and (4) Red: Overfished ($B < B_{MSY}$)

and undergoing overfishing ($F > F_{MSY}$). The Kobe plot was formulated on the basis that tuna RFMO conventions specify that the desired objective for fishery management is to maintain stocks at abundance levels that can provide MSY. Since the adoption of the Kobe plot standard, two RFMOs (ICCAT and IOTC) adopted fishery management policies with the objective of rebuilding and/or maintaining stocks in the green area of the Kobe plot with high probability. More recently, an alternative to the Kobe plot was proposed in the WCPFC, the Majuro plot (Figure 1b). In this framework, stock status is represented in terms of the spawning potential's depletion (relative to virgin stock ($B_{F=0}$ or $SSB_{F=0}$)) and fishing mortality (relative to $F_{MSY}$). The red zone represents spawning potential levels lower than the agreed upon LRP, a point at which recruitment failures are thought to become increasingly likely. The stock is considered overfished when abundance falls below the LRP. The orange region is for fishing mortality greater than $F_{MSY}$ but biomass above the LRP. This orange area is defined as not overfished but undergoing overfishing.

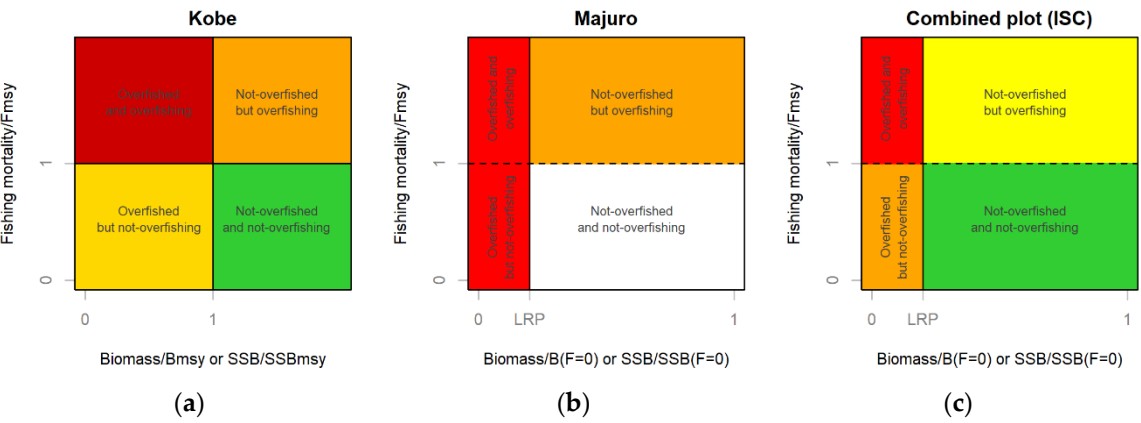

**Figure 1.** (**a**) Kobe plot, (**b**) Majuro plot, and (**c**) diagram used in the International Scientific Committee for Tuna and Tuna-like Species in the North Pacific Ocean (ISC). SSB is Spawning Stock Biomass and F is fishing mortality. MSY is Maximum Sustainable Yield.

The main difference between the two plots stems from the definition of when a stock is considered overfished. In the WCPFC, the idea behind establishing the Majuro diagram is to take appropriate action to avoid the stock being in an undesirable state or outside biological limits. When a stock is subject to overfishing ($F > F_{MSY}$), a prompt action will ensure that the stock does not breach the biological LRP, and that, ideally, stock fluctuates around the TRP [48]. In this sense, $F_{MSY}$ is used as a limit to fishing mortality. In other words, the fishing mortality will be lower than $F_{MSY}$ with high probability, and, therefore, there will be a very low probability of the stock breaching the LRP, i.e., the undesirable state will be avoided with high probability. There are different criteria to establish biological LRPs, which generally require precise knowledge of the stock–recruitment relationship (SRR). However, the most commonly applied is the 20% of the estimated value for the unexploited stock (LRP = 20% $B_{F=0}$ or LRP = 20% $SSB_{F=0}$) [49], since such a precise knowledge of the SRR is uncommon. The Kobe diagram also aims at avoiding an undesirable situation, but in this case falling below $B_{MSY}$ is considered to be the overfished state.

In addition, the International Scientific Committee for Tuna and Tuna-like Species in the North Pacific Ocean (ISC), which is responsible for providing the scientific advice for North Pacific stocks to the WCPFC and IATTC, uses a figure that combines the color scale of the Kobe plot with the stock status definition of the Majuro framework (Figure 1c). The difference of this combined plot is that it splits the overfished (B < LRP) category between those for which fishing mortality has been reduced below $F_{MSY}$ (orange) and those for which fishing mortality remains above $F_{MSY}$. Moreover, in this case, all the area above the LRP and below $F_{MSY}$ is colored in green.

The uncertainty characterized in tuna stock assessments is communicated through probability estimates of stock status and outcomes of alternative management actions. This is because different

structural and statistical assumptions are made in stock assessments, which unfold in uncertainty grids (see previous section). In general, central tendency values of model estimates (particularly stock status and RPs) are provided with deviates. The categorization of stock status in Kobe and Majuro frameworks is done using medians and in some cases a probability of each stock status option is provided (IOTC and ICCAT).

Figure 2 shows the median status of the most important tuna stocks using a common format (Kobe plot (Figure 2a), Majuro plot (Figure 2b), and the plot used in the ISC (Figure 2c)). For these figures we have used the LRP adopted by each RFMO and, when undefined, the LRP = 20% $SSB_{F=0}$ as a proxy.

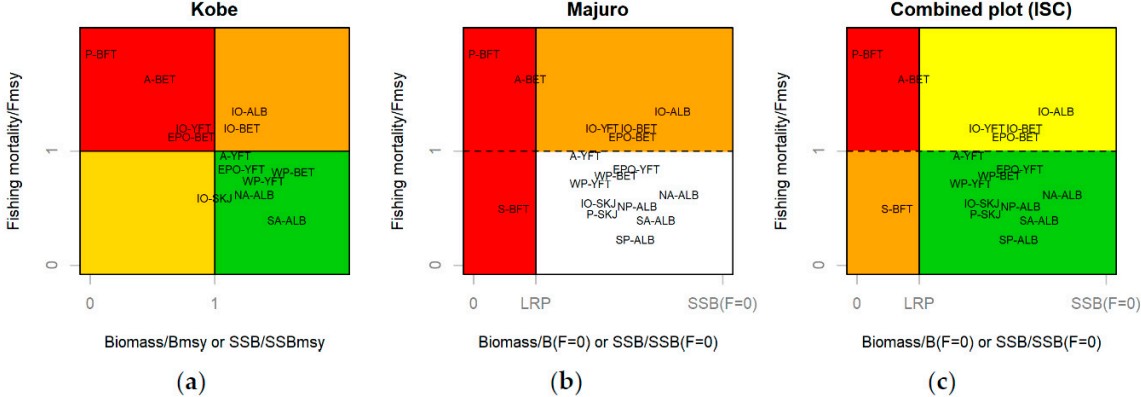

**Figure 2.** State of the most important tuna stocks using a Kobe plot (**a**), Majuro plot (**b**), and the diagram used in the ISC (**c**).

The Joint Tuna RFMO meeting, named the Kobe process, recommended the Kobe II Strategy Matrices (K2SMs) as a way to report on the probability of breaching RPs (e.g., biomass falling below or fishing mortality going over an RP) under alternative management scenarios [21]. K2SMs show the probability that stocks will remain at or above $B_{MSY}$ and at or below $F_{MSY}$ with a different probability for different catches or fishing mortality levels at different time frames.

## 4. Management of Uncertainty in Tuna RFMOs

Considering the scientific advice on stock status and management options, and, noting their own management objectives framework and existing uncertainties, RFMOs modulate the fishing activity by setting limits to the fishing effort, catch, or other type of measures. One way to manage uncertainty on fisheries under the PA is through MSE and the adoption of HSs. The five tuna RFMOs have carried out some type of MSE work, including consultation on management objectives, characterization of uncertainty of stocks' dynamics and observation, and evaluation of HSs [22]. The CCSBT has pioneered the adoption of HSs for tunas: From 2002 to 2011, the CCSBT conducted extensive work to develop an HS known as the "Bali procedure" that was adopted to help rebuild Southern bluefin by setting catch limits periodically [50]. The development of this work was initiated by a technical group of experts through specific workshops. These were scheduled to develop a work plan to focus on the specification of operating models (OMs), that represent alternative biological and fishery dynamics [23] and the evaluation of simple HS. The OMs developed in the CCSBT are the same models used to monitor the recovery of the stock (see Table 2). The HS of the CCSBT is tuned to achieve a management objective of rebuilding the stock to the interim rebuilding target point of 20% of the original spawning stock biomass by 2035 with 70% probability. The Bali procedure consists of the application of two HSs and setting catch limits from the average of the two. These are based on the trends of juvenile and adult fish indicators (CPUE and aerial survey) and the estimates of a biomass random effects model [51].

The ICCAT and IOTC have adopted HCRs for North Atlantic albacore and Indian Ocean skipjack, respectively [52,53]. The main difference with the CCSBT is that both RFMOs have adopted HCRs and

not the entire HS, which includes the specifications on the data and the stock assessment methods to be used. In both cases the adoption of the HCRs represents a first step toward adopting fully specified MPs shortly.

The MSE process is also developing fast for many other tuna stocks in the ICCAT, IOTC, WCPFC, and IATTC: The IOTC Scientific Committee started a work plan to evaluate HSs using MSE for albacore, bigeye, yellowfin, and skipjack in 2012 (and swordfish in 2018). Since then, a small ad hoc working group has been tasked to develop MSE works and to report to the IOTC through the Technical Committee on Management Procedures (TCMP) organized directly before the IOTC Annual Commission meetings. The TCMP is the formal communication forum between science and management to enhance the decision-making response of the commission in relation to the adoption of HSs [54]. The IOTC's Resolution 15-10 "*On target and limit reference points and a decision framework*" defines a series of interim target and limit RPs for albacore, bigeye, yellowfin, skipjack, and swordfish [55]. Note that these resolutions are explicitly based on IOTC's Resolution 12-01 "*On the implementation of the precautionary approach.*" Also, it is important to note that currently, despite some agreement on the management objectives, there is no time frame or probability levels agreed upon for any of these stocks. Along these lines, the IOTC called the TCMP to define the overarching management objectives to guide the development of management procedures (or HSs) for the IOTC fisheries [56]. With regards to albacore, bigeye, and yellowfin, the MSE frameworks are at very advanced stages and will be in a position to evaluate HSs in the coming years [57]. The uncertainty for these stocks is characterized through OM grids conditioned from the latest stock assessments with alternatives for natural mortality, steepness, selectivity, and dynamic catchability.

With regards to the adopted HCR for Indian Ocean SKJ, the IOTC's Resolution 16-02 indicates the procedure to be followed to establish the catch limits in three-year periods directly from stock status estimated through a stock assessment endorsed by the Scientific Committee. Resolution 16-02 specifies a relationship between stock status (spawning biomass relative to unfished levels) and fishing intensity (exploitation rate relative to target exploitation rate, Figure 3). This resolution also defines the target reference point for skipjack as $SSB_{TAR} = 40\%$ $SSB_0$ and the limit reference point at $SSB_{LIM} = 20\%$ $SSB_0$. Note that the LRP (20% $SSB_0$) is different from the "safety level" of 10% $SSB_0$ defined in the HCR. Note also that the biological TRP is the threshold of the HCR, the point below which the fishing mortality used to establish catch limits starts to decrease.

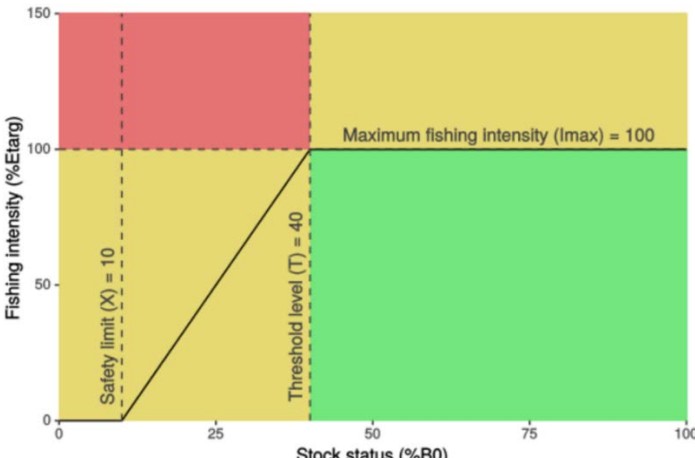

**Figure 3.** Harvest control rule adopted for Indian Ocean skipjack (IOTC, Resolution 16-02).

In ICCAT, the Standing Committee for Research and Statistics (SCRS) has fostered the development of MSE under the principles of the PA for tuna stocks in the Atlantic since 2010 [58]. This development has been followed by successive requests to evaluate HCRs consistent with ICCAT objectives and decision-making principles [59]. To date, an MSE has mostly been developed for North Atlantic albacore

and Atlantic bluefin tuna. With regards to the North Atlantic albacore MSE, this was specifically developed to support the adoption of a HCR in 2017 [52]. The MSE developed includes an HS that is similar to the latest assessment of this stock, i.e., simulating the catch data and CPUE series used for the 2016 stock assessment, together with the same model and model specifications (shape, initial biomass, and range of parameters). This work covered the uncertainty inherent to this fishery through a range of options for natural mortality, steepness, dynamic catchability, and available information [60–63]. Recently, another study used the MSE framework for North Atlantic albacore to show the benefits of HCRs to mitigate the impacts of climate change [64].

Based upon the results of the HCR evaluation for North Atlantic albacore, the ICCAT adopted a model-based HCR for this stock (Figure 4), which was used in 2017 to fix catch limits for the period 2018–2020 and in 2020 to recommend catch limits for the period 2021–2023. The PA is embedded in this recommendation as it specifies a target fishing mortality of $F_{TAR}$ = 80% $F_{MSY}$ and it has been evaluated to achieve the management objective of maintaining the stock in the green quadrant of the Kobe plot at least with 60% probability. The biomass limit reference point is set as LRP = 40% $B_{MSY}$ and the threshold reference point of the HCR is $B_{MSY}$. This HCR also considers one $F_{MIN}$ of 10% $F_{MSY}$ to ensure a level of catch for scientific monitoring [52].

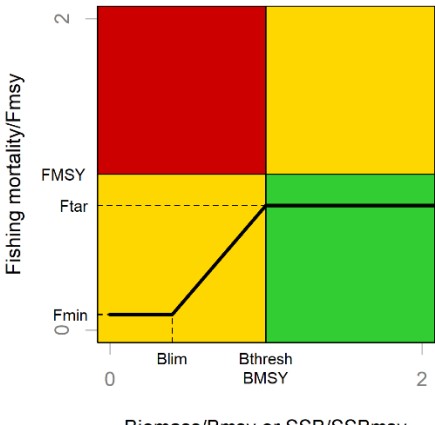

**Figure 4.** Harvest control rule adopted for North Atlantic albacore.

For Atlantic bluefin tuna, noting the high priority placed by the commission on the completion of the MSE work program and the development of new and/or improved assessment methods, the Atlantic Wide Research Programme for Bluefin Tuna (GBYP) Modelling and MSE Group was created in 2014 [65]. This group has structured a work plan in five components: 1. Data collation, management and synthesis; 2. Review and selection of alternative stock assessment models; 3. Development of the MSE modeling platform; 4. Capacity building in harvest strategies, reference points, and MSE; and 5. Consultation and engagement in design and evaluation of harvest strategies. An important step of the Atlantic bluefin tuna MSE is that it allows for mixing of stocks over several areas, in contrast with the current assumption of two separate stocks.

The Scientific Advisory Committee (SAC) has led the MSE process in the IATTC, and it has been responsible for the technical work that has guided the adoption of target and limit RPs as well as the stock assessments. In 2003, SAC organized a workshop with the aim of describing the management objectives for the stocks under its mandate and defining appropriate reference points [66]. After consultations, in 2014 the IATTC adopted an interim target and limited RPs for tropical tuna stocks (bigeye, yellowfin, and skipjack). The TRPs are the biomass and fishing mortality rate corresponding to MSY ($B_{MSY}$ and $F_{MSY}$, respectively), which have been the unofficial TRPs used in managing tuna in the Eastern Pacific Ocean through a pseudo-HCR. The LRPs are those associated with a 50% reduction in recruitment under a conservative assumption of the stock–recruitment relationship, which is based on biological grounds to protect a stock from serious, slowly reversible, or irreversible fishing impacts.

This approach has led to the LRP of 7.7% of $SSB_0$. In general, this is interpreted as ensuring that recruitment is not substantially impacted. Following the adoption of the target and limit RPs, SAC has developed MSE frameworks for evaluating more elaborated HCRs for tropical and other tunas, and this work is expected to lead to the adoption of formal HSs in the near future [67–72].

The WCPFC has also developed a work plan toward adopting HSs. The technical work has been focused on estimating the impact of different management objectives (including specific time frames, levels of risks, and probabilities of overexploited stocks) on fishery performance and has included economic principles of fisheries like the maximum economic yield as a RP and historical catch rates for South Pacific albacore, bigeye, skipjack, yellowfin, and Pacific bluefin tuna [73–80]. For that, alternative parameter sets are used to condition OM grids along similar lines to the uncertainty grids covered in the WCPFC stock assessments [80]. The technical work has been communicated through specific Management Objective Workshops (2012–2016), where assistance has been provided to the commission to understand the purpose and implications of management objectives; to understand both the role of appropriate RPs and the process of evaluating potential management measures in the achievement of management objectives; and to develop a list of recommended management objectives to guide the management of fisheries by the WCPFC [81–83]. These workshops have allowed identifying and refining potential TRPs and proposing conservation and management measures (CMMs) for establishing HSs for key tuna stocks. With regards to how the PA is dealt with in the WCPFC, overall, the proposed TRPs correspond to depletion levels well above the estimated MSY with the aim of improving the economic performance of fisheries and minimizing the probability of breaching LRPs.

## 5. Discussion and Recommendations

Uncertainty is inherent to fisheries and needs to be analyzed carefully. Overlooking the uncertainty in key biological processes and fishery dynamics can have a large impact on achieving fisheries management objectives [84] and can potentially lead to management failure [16]. In this document, we review the different ways that the RFMOs in charge of maintaining tuna stocks at sustainable levels treat uncertainty. Our work aligns with the Joint Tuna RFMOs initiative and the Kobe process [17] in the aim of standardizing stock assessment methods and communication formats toward a scientifically sound and harmonized sustainable management of tuna stocks [85,86]. For this, we have reviewed the similarities and differences between the characterization, communication, and management of uncertainty and provided suggestions for harmonizing practices.

In general, tuna stock assessment has evolved toward highly parameterized, integrated statistical modeling frameworks [87]. The use of simple models such as catch-based or biomass dynamic models can often produce an accurate diagnostic of stock status but cannot account for fishery particularities such as fishery dynamics of catch at size/age, the seasonal/spatial distribution of stocks, key biological processes, and technological development. Today, only Atlantic stocks are evaluated using catch-based or biomass dynamic models. In one case biomass dynamic models are used in combination with fully integrated models (Atlantic yellowfin). In the case of Western Atlantic bluefin, the complement to the integrated model is a size-based model. Thus, in only two of the most important tuna stocks the uncertain relationship between model inputs and outputs (model uncertainty) is characterized [9]. Simple models are often used by other RFMOs but only for exploratory purposes, and scientific advice is not built upon their results.

Best practice for characterizing structural uncertainty in stock assessments is through a "grid approach" that includes uncertainty related to model specification with options for key biological parameters and data streams [87]. Since 2020 this approach has been followed by all tuna RFMOs. The key biological processes of fish are reproduction, growth, and natural mortality, and thus, it is important to quantify the impact of the lack of knowledge about these processes for a better fisheries management [16]. Ranges of parameters or alternative values for these processes are used in the structural uncertainty grids for most tuna stocks. One of the most uncertain aspects of fish biology is the relationship between the available spawning stock and the following recruitment, the so-called

stock–recruitment relationship, which can have a large impact on the outcome of stock assessments. In stock assessment models, the amount of spawning biomass determines the expected recruitment in the following year through a functional relationship. The key parameter of the stock–recruitment models is steepness $h$, and it is considered as one of the most difficult parameters to estimate [21], thus it is a common factor in most tuna uncertainty grids. Since 2020, in all RFMOs the values have ranged between 0.65 and 1. Meta-analytical studies recommend using $h = 0.8$ as a default for tuna stocks [30]. In some RFMOs, the range of steepness values does not differ among species. For example, in the IOTC and WCPFC, the three tropical tuna stocks are characterized with the same range of values (0.7–0.9 in the IOTC and 0.65–0.95 in the WCPFC). There is evidence that values below 0.8 may be implausible for skipjack [34], and even for yellowfin, which are considered more productive than bigeye [88]. It seems insightful to harmonize the use of steepness values for stocks globally rather than using the same range for the different species within an RFMO—in other words, using the same species-specific range of values in all RFMOs. It seems reasonable that the values of steepness for albacore, bluefin, and bigeye can be lower those for skipjack and yellowfin.

Input uncertainty is also explored using alternative data and weighting. The data used in stock assessments consist of catch, abundance indices, size frequency data, and sometimes fishery-independent indices such as surveys or tagging programs. The model configurations used in the most important tuna stocks reflect that uncertainty is characterized for CPUE, tagging, and size data across tuna RFMOs. Catch is assumed known in all stock assessments that use the grid approach. However, in some cases, scenarios of catch reporting are used to characterize uncertainty. This is the case of East Atlantic bluefin tuna, where "inflated" and "reported" catch scenarios are developed as alternative views of the history of the fishery but not in a grid to be averaged across [47]. One exploratory analysis [89] finds that potential problems with catch reporting in the Indian Ocean do not have a large impact on yellowfin's stock status estimates but may be important for reference points such as the estimated MSY.

With regards to the "grid approach," it is recommended that all individual model configurations should be biologically plausible and satisfy model diagnostics of fit [87]. Because of time constraints, model diagnostics are often applied to one preferred "reference case" from the grid. Extending the plausibility and statistical tests to all model configurations is something to pursue by tuna RFMOs, for which methodologies will need to be developed. This was done in the IATTC in the latest bigeye and yellowfin stock assessments, where diagnostics of fits were used to validate or discard individual model runs from the uncertainty grids [32,33].

Statistical uncertainty is characterized in all tuna RFMOs. Until recently, exploring this uncertainty was difficult due to the time required to run highly parameterized models with stochasticity. The method applied in the recent assessments of the ICCAT and IOTC [45] is a promising way to characterize statistical uncertainty once models like SS3 or Multifan-CL are run in deterministic mode, which is significantly faster than computing the Hessian, which is normally done for a "reference case" of the grid only.

It is also important to reduce, to the greatest possible extent, the major uncertainties that undermine scientific advice in fisheries by improving knowledge in biological processes and the interaction between fishing gears and fish stocks [16]. For this, tuna RFMOs are currently undergoing important initiatives. For example, the Pacific Tuna Tagging Programme, the Regional Tagging Program in the Eastern Pacific Ocean, the CCSBT Tagging Program, and the Indian Ocean Tuna Tagging Programme have increased knowledge on movements of the most important tuna stocks and improved the scientific basis for estimation of abundance, exploitation rates, selectivity, and key biological processes like growth and natural mortality [90–93]. This information is already being used in all but Atlantic stock assessments. In ICCAT, the Atlantic-Wide Research Programme for Bluefin Tuna and the Atlantic Ocean Tropical Tuna Tagging Programme are producing large quantities of data that are expected to reduce uncertainties in the dynamics of Atlantic stocks.

Fishery data like CPUE can be problematic due to the problems in defining fishing effort, geographical coverage of fishing operations, and other limiting factors [94]. For this, it is important to include fishery-independent information to estimate abundance of tuna stocks. For example, the close-kin mark-recapture (CKMR) is an innovative approach that allows estimating abundance and biological parameters by finding pairs of related individuals in a population based on their genetic make-up [95]. The data obtained by this technique are currently part of the HS for Southern bluefin tuna in the CCSBT and its potential application to other tuna RFMOs is being explored [96]. Another source of fishery-independent data is the biomass information recorded by satellite GPS tracking echosounder buoys used by purse seine fleets that use fish aggregating devices (FADs) [97,98]. Echosounder buoys can measure the biomass of fish underneath FADs when tunas are not being fished, and, when adequately analyzed, this information can be used to develop fish abundance time series for tropical tunas. The index for Atlantic yellowfin was used in the last assessment of this stock, and it is expected that its use is generalized to other tuna RFMOs as well [99]. All these initiatives are only examples of the scientific and financial efforts dedicated by tuna RFMOs to improve stock assessments and reduce uncertainty in fisheries. These are expected to increase knowledge of tunas' biological processes and to improve the scientific advice framework of tuna RFMOs.

While aiming at reducing uncertainty, all tuna RFMOs have also agreed to incorporate uncertainty and risk in the communication of the stock assessment results [8]. In general, scientific advice on the status of the stocks and recommended management is averaged across the different scenarios of the uncertainty grids. The central tendency measure (generally median) is used to provide a category of the state of exploitation of tuna stocks using somewhat comparable but different frameworks. There are important implications for the alternative definitions of overfished for fish stocks (Figure 2). One paradigmatic example is Indian Ocean skipjack: It is currently estimated that, on average, the stock is exactly at its biomass target reference point (40% $SSB_0$) [100], which is estimated to be between 1.32–2.12 of $SSB_{MSY}$. However, the IOTC estimated that there is 49% of probability of being overfished. This is because the overfished status is defined as being above or below the adopted biomass TRP. In reality, the stock is estimated to be fluctuating around a precautionary TRP with a null probability of being below the adopted LRP, a benchmark that should not be breached with any substantial probability. However, as with the current Kobe framework the overfished status is defined as below the target, skipjack is defined as overfished with nearly 50% probability. In this regard, it is incompatible to maintain the stock fluctuating around the biological TRP with high probability while achieving a very low probability of not being estimated as overfished. Instead, we think that there is high probability that undesired risks to the stock are being avoided if a stock is around a precautionary target. For this, in cases where precautionary target and limit reference points and a harvest strategy (or HCR) are adopted we consider it would be best practice to determine overfished as biomass being below a precautionary LRP (e.g., 20% SSB0), as with the frameworks used in the WCPFC and ISC. This ramework would still prescribe management action when fishing mortality is estimated above $F_{MSY}$ or another $F_{TARGET}$, or biomass is below the target, within a HS or HCR.

The principles of Kobe and Majuro frameworks are somewhat combined in the framework used by ISC. With this framework it is possible to distinguish stocks that are cataloged as overfished but not subject to overfishing (B < $B_{LIM}$ but F < $F_{MSY}$) (e.g., southern bluefin tuna) and those that are overfished and subject to overfishing (B < $B_{LIM}$ and F > $F_{MSY}$) (e.g., Pacific bluefin) where the current fishing pressure exceeds scientific recommendation. The diagram used in the ISC can also be controversial as it would color in green the stocks that are very close to limit reference points. For this, we recommend the use of the Majuro plots for stocks with adopted precautionary reference points and a harvest strategy (or HCR) (Figure 1b). A potential improvement would be to assign a new color (pale red or pink) to stocks that are overfished (B < $B_{LRP}$) but not subject to overfishing (F < $F_{MSY}$).

In cases where HSs or HCRs are not adopted, management advice can be supported by projections of the stock assessment models under alternative catch or fishing mortality levels. The results of these projections are communicated using Kobe II Strategy Matrices, which are the agreed way to report the

probability of something happening (e.g., biomass falling below $B_{MSY}$ or fishing mortality going over $F_{MSY}$) under alternative management scenarios [21]. The probabilistic results shown in the Kobe plots and K2SMs are calculated from all the models, scenarios, and iterations agreed upon to represent the plausible dynamics of the fishery for the provision of management advice. With the gradual move toward HSs, this form of communication of advice will be replaced by a direct recommendation for limits of catch or effort (or any other measure).

The Kobe process has contributed to the adoption of common frameworks that embrace the precautionary approach for management in tuna RFMOs. All tuna RFMOs are committed to a path of adopting HSs for the most important stocks to address uncertainty in fisheries and to achieve management goals. All evaluations of HS to facilitate adoption are being carried out using MSE. This process has been accelerated by the conditions imposed by the Marine Stewardship Council (MSC) for the certification of tuna fisheries. Under the Joint Tuna RFMO framework, a common development of MSE has been pursued and periodically scientists from the five RFMOs share views and methodologies [101]. Today, only the CCSBT has adopted a fully specified HS to pilot the recovery of Southern bluefin tuna. The IOTC and ICCAT have adopted HCRs and have defined management objectives and RPs for Indian Ocean skipjack and North Atlantic albacore stocks, respectively. The main difference between the case of the CCSBT and the latter two is that in the first, the complete HS (including data, data analysis, and decision process) are prespecified and for the others, what it is agreed upon is the course of action (e.g., setting catch limits) once the results of the assessment are endorsed by their scientific committees. The state of development of MSE is different across stocks, but overall there are roadmaps to develop MSEs to support the adoption of HSs and HCRs in all tuna RFMOs for the most important tuna stocks in the next 3–5 years.

**Author Contributions:** All authors have contributed to preparation of this work. Conceptualization, G.M., H.M., J.S., V.R. and H.A.; formal analysis and investigation, G.M. and H.M.; writing and original draft preparation, G.M. and H.M.; writing, review and editing, G.M., H.M., J.S., V.R. and H.A. All authors have all read and agreed to the published version of the manuscript.

**Funding:** This study was funded by the International Seafood Sustainability Foundation (ISSF). The report and its results, professional opinions, and conclusions are solely the work of the authors.

**Acknowledgments:** We acknowledge ISSF for hosting the 2018 stock assessment workshop: "Review of current t-RFMO practice in stock status determination" [87] in which the topics in this manuscript were discussed, and thank the participants in the workshop for their contributions. This paper is contribution n° 999 from AZTI, Marine Research, Basque Research and Technology Alliance (BRTA).

**Conflicts of Interest:** The authors declare no conflict of interest. The funders had no role in the design of the study; in the collection, analyses, or interpretation of data; in the writing of the manuscript, or in the decision to publish the results.

## Appendix A

- Catch-based models: Relatively simple methods to obtain plausible MSY estimates and other biological parameters from catch data, based on assumptions on resilience (corresponding to the intrinsic growth rate r in the surplus production model) and the plausible range of relative stock sizes at the beginning of the time series. The algorithm by Martell and Froese (2012) has been validated against analytical fish stock assessment estimates of MSY. Good agreement was found between stock assessment MSY estimates and the geometric mean of MSY values calculated from the plausible r-K pairs [102]. A catch-based approach relies on the assumption that catch reflects fish abundance and productivity. This principle is controversial, especially when management interventions change through the history of catch time series. However, catch-based methods are widely used to assess data-poor fisheries and to produce large scale overviews of the state of fisheries [103].
- ASPIC ([104]) is a non-equilibrium implementation of the well-known surplus production model of Schaefer [105,106]. ASPIC also fits the generalized stock production model of Pella and Tomlinson (1969). ASPIC can fit data from up to 10 data series of fishery-dependent or fishery-independent

indices, and uses bootstrapping to construct approximate nonparametric confidence intervals and to correct for bias. In addition, ASPIC can fit the model by varying the relative importance placed on yield versus measures of effort or indices of abundance. The model has been extensively reviewed and tested in the context of various applications to tuna stocks via the ICCAT by Prager [104,107]. Because of its limited data requirements, this model is easy to use and many national scientists are familiar with it. ASPIC is fast to run and facilitates simulation testing. Because of the limited data requirements, it allows the use of longer time series when data from earlier periods are usually poor. It only estimates a few parameters but these are typically the ones needed to provide management advice and estimate RPs. ASPIC quickly produces diagnostics, bootstrap results, and projections.

- mpb is an R package for running and simulation testing biomass-based stock assessment models. The package is part of FLR [108], a suite of open source R packages that are extensible and able to interact with many R packages. It has methods for plotting, examining goodness of fit, estimating uncertainly, deriving quantities used to provide management advice, running projections, simulating harvest control rules (HCRs), and for conducting management strategy evaluation (MSE) [27].

- JABBA (Just Another Bayesian Biomass Assessment) is a generalized Bayesian state-space surplus production model framework [109]. JABBA is coded within a user-friendly R interface to provide a means to generate reproducible stock status estimates and diagnostics. JABBA is generalized in the sense that the production function can take on various forms, including conventional Fox and Schaefer production functions, which can be fit based on a range of alternative error assumptions. The model is formulated to accommodate multiple CPUE series for fisheries. The assessment input data can comprise multiple, partially conflicting, fisheries-dependent abundance indices over varying time spans, as commonly encountered in assessments of large pelagic fishes. The inbuilt fit diagnostics can be applied to identify conflicting abundance indices toward selecting candidate base-case scenarios. JABBA can be used to produce a large number of alternative scenarios, including readily presentable diagnostic and output graphs.

- Virtual population analysis (VPA) methods have been widely used by the SCRS for stock assessment purposes, with arguably fewer assumptions than biomass dynamic approaches. VPA can handle varying selectivity and, in general, projections can accommodate some of the management issues (size limits, etc.). It can accommodate multiple CPUE indices with different selectivity. The method can only estimate uncertainty within the model through bootstrapping; assumed catch at age (CAA) is known without error.

- Multifan-CL is a sophisticated computer program that implements a statistical, length-based, age-structured model for use in fisheries stock assessment [25]. Multifan-CL provides a statistically-based, robust method of length–frequency analysis. Multifan-CL is now used routinely for tuna stock assessments by the Oceanic Fisheries Programme (OFP) of the Secretariat of the Pacific Community (SPC) in the Western and Central Pacific Ocean (WCPO).

- Stock synthesis (SS) is a fully integrated age-structured statistical model [24]. The structure of stock synthesis allows for building simple to complex models depending upon the data available. As a result, the SS modeling framework is designed to allow the user to control the majority of the assumptions that go into the model. SS assumes that the observational data are a random and unbiased sample of the fishery and/or survey they are intended to represent. The overall model contains subcomponents which simulate the population dynamics of the stock and fisheries, derive the expected values for the various observed data, and quantify the magnitude of difference between observed and expected data. Stock synthesis provides a statistical framework for calibration of a population dynamic model using a diversity of fishery and survey data. SS is most flexible in its ability to utilize a wide diversity of age, size, and aggregate data from fisheries and surveys. It is designed to accommodate both age and size structure in the population and with multiple stock sub-areas. Selectivity can be cast as age-specific only,

size-specific in the observations only, or size-specific with the ability to capture the major effect of size-specific survivorship. While SS can accommodate a multitude of data types, at least a catch time series and an index of abundance are required. Conversely, a model can be built that incorporates multiple areas, seasons, sexes, growth and growth morphologies, as well as tagging data. Environmental data can also be used to modulate the parameters of the model. Size and age structure, size-at-age, aging error and bias, and sex ratio can also be incorporated. The SS model output is commensurate with the complexity of the model configuration and observational data. All estimated parameters are output with standard deviations. Derived quantities include typical management benchmarks such as MSY, $F_{MSY}$, and $B_{MSY}$, and Spawners per Recruit. Typical matrices of numbers-at-age, growth, age–length keys are also provided.

- Operating Model developed for SBT MP testing (SBT-OM): The performance of the management procedure currently in place for Southern bluefin tuna is evaluated using a specifically tailored age-structured model [37,110]. The model allows for historical trends in growth. It assumes a Beverton–Holt recruitment function with log-normal auto-correlated errors. The relationship includes a parameter that allows for depensatory effects. The model assumes fishing for each of the fisheries considered as a pulse that takes place in one or two fishing seasons. The length–age relationship and fishery-specific length–weight relationships are considered known, specified by a time-varying growth schedule estimated outside the model. The model can accommodate tag return data that can be used to provide estimates of fishing mortality and natural mortality. The model also uses CPUE indices as an aggregate index and uses aerial survey data as a relative index of biomass of ages 2–4. Other indices can also be incorporated to the model. The model also uses close-kin data, which considers juvenile and adult individuals and is used to help the model estimate parent–offspring pairs.

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
