# Peer review of "Characterization, Communication, and Management of Uncertainty in Tuna Fisheries"

_sustainability, doi:10.3390/su12198245_

Round 1

Reviewer 1 Report

This paper is about the way the five tuna RFMOs manage uncertainty. The authors say management of uncertainty or risk is really important if effective measure are to be put in place by RFMOs to maintain the sustainability of global tuna stocks. The paper explains that currently there are differences in the ways the RFMOs deal with uncertainty, but they have all signed up to the so-called Kobe process, whereby they are committed to adopting ‘Harvest Strategies’ supported by a ‘Management Strategy Evaluation’ (MSE) to assess the consequences of uncertainty for attaining sustainable tuna fisheries. The authors use the paper to describe the ways the five tuna RFMOs are using roadmaps to develop MSEs to achieve common Harvest Strategies within the next five years.

In my view, this paper deserves to be published virtually as it stands, because it is an impressive analysis of how a major problem (how to deal with uncertainty) is being handled by the RFMOs who manage the most valuable global fishery. I have only two reservations. First, the authors could clarify the relationship between their three kinds of error (observation error; model error; and process error) and their three kinds of uncertainty (model uncertainty; input uncertainty; and statistical uncertainty). Second, the authors could tell us (a) whether the Kobe process with its MSE has been fully adopted in other fisheries, and if so with what effect; and (b) how effective the partial adoption of the Kobe process by each of the five tuna RFMOs has been so far in improving the management of the stocks for which they are responsible, and what lessons we should draw from their respective experiences. 

Author Response

Thanks for the positive feedback. We respond to each of the questions below.

I have only two reservations.

  • First, the authors could clarify the relationship between their three kinds of error (observation error; model error; and process error) and their three kinds of uncertainty (model uncertainty; input uncertainty; and statistical uncertainty).

AU: We have tried to clarify this (also following the main concern of Reviewer 2). It should be clearer now.

  • Second, the authors could tell us (a) whether the Kobe process with its MSE has been fully adopted in other fisheries, and if so with what effect;

AU: The Kobe process is not fully developed yet. However, the process of adopting HCR/MPs for the most relevant stocks is common to all tuna RFMOs. This includes the adoption of a HCR for North Atlantic albacore (in 2017) and Indian Ocean skipjack (in 2016). Technical works are in progress for many other stocks worldwide too.

(b) how effective the partial adoption of the Kobe process by each of the five tuna RFMOs has been so far in improving the management of the stocks for which they are responsible, and what lessons we should draw from their respective experiences. 

AU: The Kobe process is being effective because it has speeded up the adoption of Reference Points (Target and Limits) and HCR/MP across tuna RFMOs. Also, these are built upon the Precautionary Approach. Finally, tuna RFMOs are including ecosystem aspects in their mandates or work priorities, some have adopted more regulations for a better management, for mitigating bycatch, etc...

Thanks for the suggestions. We have added one sentence in the last paragraph of the discussion that should help clarify this.

Reviewer 2 Report

The word file is attached below. 

Author Response

Thanks for the positive feedback. We have responded to each of the minor issues one by one:

-The way for classification of uncertainties is a little confusing. I have listed up words on uncertainties and errors in the order appeared in this paper. It is difficult to catch up relations among the uncertainty words. The categorization of uncertainties in fisheries assessment is sometimes vague even in the referring papers, but it is better to use consistent words through the text.

Lines64-65; “systemic uncertainty”/ “aleatoric or statistical uncertainty”. We have reworded to “systemic or structural uncertainty” to avoid confusion. This is because in modelling outside fisheries the term “systemic” is used while in fisheries it is more often referred as “structural”. This may be clearer now.

AU: We have tried to clarify this (also following the main concern of Reviewer 1). It should be clearer now.

Lines81-84; “observation errors”/ “model errors”/ “process errors”.

AU: This is correct.

Lines147, 149, 167, 212; “systemic or structural uncertainty” can be due to “model errors (model uncertainty)” and “observation errors (input uncertainty)”/ “statistical uncertainty (process errors)”

AU: We have simplified the text a little bit. Systemic or structural uncertainty is the result of model uncertainty and input uncertainty. Then, each of those is explained below. We have removed the reference to errors to avoid confusion. We have also added the numbering (2) to statistical uncertainty. It is probably clearer now, thanks.

-Line 387; “GBYP” There is no explanation of GBYP.

AU: The full name of the project has been added.

-Lines541-542; “In reality, the stock is estimated to be fluctuating around a precautionary TRP with a null probability of being below the adopted LRP, a benchmark that should not be exceeded with any substantial probability”. The meaning of this sentence is difficult to be understood. Is the LRP (biomass-RP?) should not be exceeded?

AU: We have re-worded using “breached” to make it more readable.

-Line 555; “not being overfished(e.g. southern bluefin tuna)” and “being overfished (e.g. Pacific bluefin and Atlantic bigeye)” Is it appropriate for the words of “being overfished” to be replaced with “being overfishing”? This is because usually we call “overfishing” when F>Fmsy while “overfished” when B<Bmsy.

AU: We have changed the labels of the quadrants in Figure 1 and also add the term “subject to overfishing” in this part of the manuscript.

- Line 637; “Biodyn” does not appear in this paper, but “mpb” does appear in the main text and Table 1 . Please sort out package names around FLR.

AU: Yes, this was an error. The package was used to be named as “biodyn” but now is officially named as “mpb”. We have made sure that the word biodyn does not appear anywhere in the text. The name in the Appendix is now mpb.

-Line 655 (VPA) and 689 (Operating model developed for SBT MP testing); References are needed for those explanations.

AU: Two references to the Report of the CCSBT Scientific Committee and the working document have been added.

-Table 1; It is better to cite Appendix A (explanation of each column of stock assessment method) in the table caption.

AU: This text has been added to the caption: “A brief summary of each model is provided in Appendix A.”

-Table 2; There is no explanation of abbreviations such as “WA --“ and “A --“. AU: This has been explained with text in the caption “WA stands for Western Atlantic, A for Atlantic, IO for Indian Ocean, P for Pacific, SP for Southern Pacific and EPO for Eastern Pacific Ocean.”

-In addition, Pacific bluefin tuna and North Pacific albacore are not in Table 2 while those stock assessment models are also fully integrated (SS).

AU: Yes, and there is a reason for that. In the most recent assessments of these stocks a single base case was used to estimate stock status and therefore, the systemic uncertainty was explored in a reference grid. Instead, the impact of biological parameters and other model choices was explored using sensitivity analyses. A brief note has been added to clarify this: “Also note that the stock assessments of Pacific bluefin (IATTC-ISC) and North Pacific albacore (WCPFC-ISC) do not use uncertainty grids to evaluate stock status and therefore, they are not included in Table 2.